# High Intrinsic Oncogenic Potential in the Myc-Box-Deficient *Hydra* Myc3 Protein

**DOI:** 10.3390/cells12091265

**Published:** 2023-04-26

**Authors:** Marion Lechable, Xuechen Tang, Stefan Siebert, Angelika Feldbacher, Monica L. Fernández-Quintero, Kathrin Breuker, Celina E. Juliano, Klaus R. Liedl, Bert Hobmayer, Markus Hartl

**Affiliations:** 1Institute of Zoology, University of Innsbruck, 6020 Innsbruck, Austria; 2Center for Molecular Biosciences Innsbruck (CMBI), University of Innsbruck, 6020 Innsbruck, Austria; 3Institute of Inorganic and Theoretical Chemistry, University of Innsbruck, 6020 Innsbruck, Austria; 4Department of Molecular and Cellular Biology, University of California, Davis, CA 95616, USA; 5Institute of Organic Chemistry, University of Innsbruck, 6020 Innsbruck, Austria; 6Institute of Biochemistry, University of Innsbruck, 6020 Innsbruck, Austria

**Keywords:** cnidaria, development, gene regulation, signal transduction, interstitial stem cell, neurogenesis, oncogene, cancer

## Abstract

The proto-oncogene *myc* has been intensively studied primarily in vertebrate cell culture systems. Myc transcription factors control fundamental cellular processes such as cell proliferation, cell cycle control and stem cell maintenance. Myc interacts with the Max protein and Myc/Max heterodimers regulate thousands of target genes. The genome of the freshwater polyp *Hydra* encodes four *myc* genes (*myc1-4*). Previous structural and biochemical characterization showed that the *Hydra* Myc1 and Myc2 proteins share high similarities with vertebrate c-Myc, and their expression patterns suggested a function in adult stem cell maintenance. In contrast, an additional *Hydra* Myc protein termed Myc3 is highly divergent, lacking the common N-terminal domain and all conserved Myc-boxes. Single cell transcriptome analysis revealed that the *myc3* gene is expressed in a distinct population of interstitial precursor cells committed to nerve- and gland-cell differentiation, where the Myc3 protein may counteract the stemness actions of Myc1 and Myc2 and thereby allow the implementation of a differentiation program. In vitro DNA binding studies showed that Myc3 dimerizes with *Hydra* Max, and this dimer efficiently binds to DNA containing the canonical Myc consensus motif (E-box). In vivo cell transformation assays in avian fibroblast cultures further revealed an unexpected high potential for oncogenic transformation in the conserved Myc3 C-terminus, as compared to *Hydra* Myc2 or Myc1. Structure modeling of the Myc3 protein predicted conserved amino acid residues in its bHLH-LZ domain engaged in Myc3/Max dimerization. Mutating these amino acid residues in the human c-Myc (MYC) sequence resulted in a significant decrease in its cell transformation potential. We discuss our findings in the context of oncogenic transformation and cell differentiation, both relevant for human cancer, where Myc represents a major driver.

## 1. Introduction

The *myc* gene was originally identified as the transforming principle (v-*myc*) in the genome of the avian acute leukemia virus MC29 encoding a single hybrid protein composed of partial structural (Gag) and Myc sequences [1,2]. The oncogenic v-*myc* allele is derived from the cellular chicken c-*myc* protooncogene by retroviral transduction [3,4,5,6], and homologs of c-*myc* have been identified in all vertebrate genomes. The encoded c-Myc protein represents the key component of a transcriptional network controlling the expression of a large fraction of all human genes, thereby regulating fundamental cellular processes required for metabolism, homeostasis, growth, proliferation, differentiation, or apoptosis [5,7]. Human c-Myc (MYC) is also one of the most frequently deregulated oncoproteins in many cancer types and a hallmark of the majority of human cancers [8,9]. c-Myc and its paralogs N-Myc (MYCN) and L-Myc (MYCL) are bHLH-LZ proteins encompassing a protein dimerization domain (helix-loop-helix, leucine zipper), and a DNA contact surface (basic region) both located in the protein’s C-terminus. The N-terminus of Myc contains multiple conserved regions termed Myc boxes (MB), which interact with multiple proteins mediating crucial cellular functions including transcriptional transactivation and DNA replication [5,10,11]. Myc proteins form heterodimers with the Myc-associated factor X (Max) typically binding to a canonical DNA sequence element termed E-box (5′-CACGTG-3′) [12] in the regulatory regions of multiple Myc target genes, and thereby they form a central hub in a global gene expression network [5,13,14,15].

The Myc and Max proteins are evolutionary conserved, meaning that their amino acid sequences have a high similarity in different species. Invertebrate orthologs have initially been characterized in the triploblastic bilaterian organism *Drosophila melanogaster* [16], where dMyc and dMax bind to a large number of E-boxes and regulate the expression of many genes including key regulators of cell growth, cell size, and ribosome biogenesis [17]. Structural and functional homologs are also present in early, pre-bilaterian metazoans and pre-metazoans [5,18,19]. Among basal metazoan organisms, the diploblastic cnidarian *Hydra* has been used to study Myc and Max. Two *Hydra* c-Myc-like proteins, Myc1 and Myc2, display interaction with Max and DNA similar to vertebrate Myc proteins, and both show a basic level of oncogenic potential [20]. Thus, the principal design and basic biochemical properties of the ancestral *Hydra* Myc and Max proteins seem to be very similar to those of their vertebrate derivatives, suggesting that the principal functions of the Myc master regulator arose very early in metazoan evolution, at least 550 million years ago.

*Hydra* has a long history as a simple animal model for studying regeneration, pattern formation and stem cell biology [21,22,23,24,25]. In laboratory culture, *Hydra* polyps permanently grow and reproduce by asexual budding. Polyps exhibit a single oral-aboral body axis with tentacles and a mouth opening at one end and a foot at the opposite end. The polyp body is built by only three adult cell lineages, ectodermal and endodermal epithelial cells and interstitial cells [24,25]. All three lineages are maintained by their own pools of stem cells. The interstitial cell lineage is maintained by large numbers of multipotent interstitial stem cells, which continuously proliferate in order to self-renew and to differentiate neurons, gland cells and nematocytes, as well as gametes during rare events of sexual reproduction [24,25]. The two *Hydra* c-*myc*-like homologs, *myc1* and *myc2*, are transcriptionally activated in the interstitial cell lineage [18,20,25]. *myc1* and *myc2* are expressed in proliferating interstitial stem cells and in early proliferating precursor stages during nematocyte differentiation. Furthermore, *myc2* expression also occurs in proliferating epithelial stem cells throughout the gastric region. In addition, *myc2* is activated in cycling precursor cells during early oogenesis and spermatogenesis and in cycling nematoblast nests and gland cells, suggesting that the Myc2 protein has a possible non-redundant function in cell cycle progression and thereby contributes to stem cell maintenance [25].

In total, the *Hydra* genome sequence contains four *myc* or *myc*-like genes [20]. Whereas *myc1* and *myc2* encode for prototypical Myc proteins displaying a highly conserved C-terminal bHLH-LZ DNA binding domain and most of the Myc-boxes in the larger N-terminal transactivation domain, the other two *myc*-like genes, *myc3* and *myc4*, are more divergent. Although the predicted C-terminal bHLH-LZ domains are Myc-specific, their predicted N-terminal domains are shorter and almost completely lack Myc boxes [20]. Here, we report the detailed characterization of the highly divergent *Hydra myc* paralogue termed *myc3*. Dynamic structure predictions of the Myc3 C-terminal region and avian cell transformation assays provide evidence for the impact of specific amino acid residues on Max interaction and oncogenic potential. Furthermore, the *myc3* expression pattern in *Hydra* suggests a distinct role in cell differentiation, where the encoded Myc3 protein may act as a competitor to Myc2 or Myc1 for Max binding. We discuss these results with respect to models of cell transformation and differentiation.

## 2. Materials and Methods

### 2.1. Animals

*Hydra vulgaris* strains AEP and 105 (formerly *Hydra magnipapillata* strain 105) were used in this study. Mass cultures were kept as described [26]. Experimental animals were collected 24 h after the last feeding.

### 2.2. Whole Mount In Situ Hybridization

In situ hybridization with digoxigenin-labeled RNA probes was carried out according to protocol as described in [27], using a *myc3*-specific cDNA probe as described in [18,25].

### 2.3. Single-Cell Transcriptome Analyses

*Hydra* scRNA-seq data [28] were used to analyze the expression of Myc variants within interstitial cells. Recent UMAP representations [29] were used to visualize gene expression. The expression of *Clytia hemisphaerica myc* paralogs was analyzed using published scRNA-seq data [30].

### 2.4. Cells and Retroviruses

The constructs pRCAS-hymyc1/v-myc, pRCAS-hymyc2/v-myc, and pRCAS-v-myc have been described [18,25]. To construct the plasmids pA-hymyc3/v-myc and pA-v-myc/hymyc3, encoding hybrid proteins of *Hydra* Myc3 and v-Myc (HyMyc3/v-Myc and v-Myc/hyMyc3), the corresponding *Hydra myc3* and v-*myc*-specific segments containing overlapping sequences were amplified in four different PCRs using *Hydra myc3* cDNA and the adaptor plasmid pA-v-myc [18] as templates and the primer pairs 5′-GCATCGATACCGACCACCATGATGTATGGGCAAAGT-3′/5′-GTGCGTTCGCCTCTTGTCTACTGGCTCTTGTTTAAT-3′ (hymyc3_a), 5′-ATTAAACAAGAGCCAGTAGACAAGAGGCGAACGCAC-3′/5′-GCATCGATTAGAGGATCCCTATGCACGAGAGTTCCT-3′ (v-myc_b), 5′-GCGAATTCGCCGACCACCATGCCGCTCAGCGCCAGC-3′/5′-ATGTGTGGTTCGGCTTAAGTTCTCCTCTGAGTCTAA-3′ (v-myc_a), 5′-TTAGACTCAGAGGAGAACTTAAGCCGAACCACACAT-3′/5′-GCAAGCTTTAGAGGATCCCTAATATAACCCTTTTTT-3′ (hymyc3_b), respectively. In two subsequent PCRs, diluted pools from pairs of the first PCRs were employed as templates (hymyc3_a + v-myc_b and v-myc_a + hymyc3_b) using the relevant external primer pairs in each case, as described in [25]. The resulting PCR products were digested with *Cla*I (hymyc3/v-myc) or *Eco*RI/*Sal*I (v-myc/hymyc3), and inserted into the adaptor plasmids pA-CLA12NCO or pA-CLA12, which had been opened with *Cla*I or *Eco*RI/*Sal*I, respectively. The inserts of pA-hymyc3/v-myc, and pA-v-myc/hymyc3 were then released with *Cla*I and inserted into the retroviral RCAS-BP vector as described in [18]. To construct the plasmid pRCAS(A)BP-HA-hymyc3, fragments encompassing the coding sequences of the hemagglutinin (HA) tag (112 bp) and of *Hydra myc3* (658 bp) were amplified using pRCAS(A)BP-HA-MYC [31] or the *Hydra* myc3 cDNA as templates, using the primer pairs 5′-CTCGCGTACCACTGTGGCATCGATTCTAGACCACTGTG-3′/5′-GCCCATACATAGCGTAATCTGGAACATC-3′ and 5′-AGATTACGCTATGTATGGGCAAAGTGCG-3′/5′-ATCTGGCCCGTACATCGCATCGATAAGCTTGGGCTGCAG-3′, respectively. The PCR products were then inserted into the *Cla*I-linearized pRCAS-BP vector via Gibson assembly following the NEBuilding protocol (New England Biolabs, Ipswich, MA, USA), and the generated constructs were verified by DNA sequencing upon plasmid DNA isolation. The theoretical molecular weights (*M*_r_) of the proteins encoded by the applied RCAS constructs are: 24,378 (HA-hyMyc3); 23,413 (hyMyc3/v-Myc); 45,977 (v-Myc/hyMyc3); 46,432 (v-Myc/hyMyc2); 47,210 (v-Myc/hyMyc1); and 46,095 (v-Myc w/o Gag).

For construction of the expression plasmids pRCAS(A)BP-HA-c-myc-QN and pRCAS(A)BP-HA-c-myc-TIQN encoding human HA-tagged MYC proteins with the mutations R423Q/R424N (QN) and E410T/E417I/R423Q/R424N (TIQN), the Q5 site-directed mutagenesis protocol NEBaseChanger (New England Biolabs, Ipswich, MA, USA) was applied with the pRCAS(A)BP-HA-MYC plasmid [31] as a template. Using the mutagenesis primer pair 5′-TTGCGGAAACAAAACGAACAGTTGAAACACAAAC-3′/5′-CAAGTCCTCTTCAGAAATG-3′ for the QN mutant, and the primer pair 5′-CGACTTGTTGCGGAAACAAAACGAACAGTTGAAACACAAACTTGAAC-3′/5′-ATTTCAGAAATGAGCTTTTGCGTCTCTGCTTGGACGGACAG-3′ for the TIQN mutant, the relevant DNA segments were PCR amplified and then treated with a kinase/ligase/*Dpn*I mixture, followed by transformation into competent *Escherichia coli* XL10 gold. Successful mutagenesis was confirmed by DNA sequencing of the isolated plasmid DNA.

Cultivation of quail embryo fibroblasts (QEF), calcium phosphate-mediated DNA transfection, and cell transformation assays (focus and colony formation) were performed as described [18,25]. Proliferation of cells was monitored in real time by using the live cell imaging system IncuCyte S3 (Essen Bioscience/Sartorius, Vienna, Austria). Each aliquot of 1.25 × 10^5^ cells were seeded in an MP24-well dish (Corning, Vienna, Austria), and monitored for 72 h using phase contrast imaging every 8 h from 9 separate regions per well, using a 10× objective. To verify the integrity of the integrated proviruses in cells transfected with the constructs RCAS-vmyc/hymyc3 and RCAS-hymyc3/v-myc, genomic DNA was isolated from each group of 1 × 10^6^ cells after seven cell passages, using the Monarch genomic DNA purification kit (New England Biolabs, Ipswich, MA, USA). Each 1 µg of genomic DNA was used as a template for a standard PCR, using the RCAS-specific primer pair 5′-TGAGCTGACTCTGCTGGTG-3′/5′-GGCCCGTACATCGCATCGAT-3′, followed by direct DNA sequencing of the resulting PCR products.

### 2.5. Protein Expression and Purification

To construct prokaryotic expression plasmids encoding the bHLH-LZ regions of *Hydra* Max, and Myc3 with carboxy-terminally fused Histidin-tags (His_6_), the relevant coding regions were amplified by PCR using the primer pairs 5′-CTTTAAGAAGGAGATATACAATGGCTGATAAAAGAGCTC-3′/5′-AGTGGTGGTGGTGGTGGTGCTCTAGTGTTAGGTTTCCAC-3′ (*max*), 5′-CTTTAAGAAGGAGATATACAATGCAAGAGCCAGTATTAAG-3′/5′-AGTGGTGGTGGTGGTGGTGCAGATATAACCCTTTTTTAAGGTC-3′ (*myc3*), and then inserted into the pET21a vector providing the initiating methionine codon and a C-terminal (HIS)_6_-tag consisting of six consecutive histidine residues followed by a stop codon. The pET21a vector was opened with the restriction enzymes *Nde*I/*Xho*I followed by insertion of the PCR fragments via Gibson assembly, according to the NEBuilding protocol (New England Biolabs, Ipswich, MA, USA). The generated plasmids pET21a-hymax_bHLH-LZ-HIS and pET21a-hymyc3_bHLH-LZ-HIS encode the fusion proteins HyMax-bHLH-LZ-HIS (*M*_r_ = 11,836.27, pI = 9.89) and HyMyc3-bHLH-LZ-HIS (*M*_r_ = 11,839.82, pI = 10.13), respectively. After verification of the constructs by DNA sequencing, DNA aliquots were transformed into the *Escherichia coli* strain Rosetta (DE3) pLysS (Novagen/Merck, Darmstadt, Germany). To express the recombinant proteins, bacteria from single colonies were incubated overnight at 37 °C with shaking at 200 rpm in 20 mL of LB medium containing 100 μg/mL ampicillin and 25 μg/mL chloramphenicol. The bacteria were then transferred into 180 mL LB medium containing the same antibiotics as above, and grown at 37 °C with shaking at 200 rpm to an optical density of 0.5 (600 nm). To induce recombinant protein expression, isopropyl-β-D-thiogalactopyranoside (IPTG) was added to a final concentration of 1 mM and bacteria were incubated as above for 4 h. The bacteria were pelleted at 6000× *g* for 15 min at 4 °C, resuspended in 5 mL of column loading buffer (50 mM sodium phosphate pH 8.0, 300 mM NaCl, 10 mM imidazole) supplemented with protease inhibitors (2 µg/mL aprotinin, 1 µg/µL leupeptin, 1 µg/µL pepstatin), and then frozen at −80 °C for at least 20 min. To the thawed cell suspension, 5 µL of 1 M MgCl_2_, 50 µL of *DNAse*I (1 mg/mL), and 50 µL of lysozyme (10 mg/mL) were added, followed by incubation on ice for 60 min. The lysates were then centrifuged at 12,000× *g* for 20 min at 4 °C, and the supernatants applied for protein purification by metal-chelate affinity chromatography. Each 600 µL of the clarified lysates was loaded onto Ni-NTA spin kit 50 (Qiagen, Venlo, Netherlands) columns equilibrated with column loading buffer. Samples were centrifuged at 270× *g* for 10 min, and the bound proteins washed three times (890× *g* for 2 min) with column washing buffer (50 mM sodium phosphate pH 8.0, 300 mM NaCl, 20 mM imidazole). The His-tagged proteins were then eluted by centrifugation (890× *g* for 2 min) using column elution buffer (50 mM sodium phosphate pH 8.0, 300 mM NaCl, 500 mM imidazole) and quantified by Nanodrop photometry (PeqLab/Avantor, Radnor, PA, USA) and SDS-PAGE. The proteins were stored in aliquots at −80 °C.

### 2.6. Protein Analysis

SDS-PAGE and immunoblotting were carried out as described in [25,32]. Specific rabbit antisera recognizing v-Myc (anti-Myc-CT, anti-Myc-NT) have been described in [25]. Mouse antibodies directed against anti-tubulin (TUBA) and the HA-tag have been described in [33]. The monoclonal anti-GAPDH (#AB8245, Abcam, Cambridge, UK) was applied in a 1:1000 dilution.

### 2.7. Protein–DNA Interaction Analysis

Electrophoretic mobility shift assay (EMSA) analysis, radioactive ^32^P-labeling of the DNA probe, and signal quantification were performed as described in [18]. To generate a double-stranded DNA probe, each two complementary oligodeoxynucleotides were annealed containing either the canonical Myc binding site (E-box) from the *Hydra myc2* promoter (5′-ATAGCTCACGTGTCAATA-3′) [25,34], or in the context of an upstream stimulatory factor binding site (E-box USF), as described in [18]. DNA binding reactions (20 μL) were performed at 25 °C for 45 min in a buffer containing 10 mM Tris HCl pH 7.5, 0.5 mM EDTA, 65 mM KCl, 5 mM MgCl_2_, 1 mM DTT, 100 μg/mL BSA, and 10% (vol/vol) glycerol. Protein–DNA complexes were resolved by native 6% (wt/vol) PAGE, and radioactive signals were quantified by using a PhosphorImager and the program Image-Quant TL (GE Healthcare, Chicago, IL, USA), as described in [25].

### 2.8. Protein Structure and Homology Modeling

Homology models of the *Hydra* Myc/Max/E-box binding complexes were built and optimized based on the crystal Myc/Max/E-box structure [35] (PDB accession code: 1NKP) using MOE (Molecular Operating Environment, 2022.02 Chemical Computing Group ULC, Montreal, QC, Canada). The electrostatic properties of the *Hydra* Myc3/Max/E-box binding interface were calculated based on the respective charges and mapped on the surface of the heterodimer structure. The resulting structure models were overlaid to facilitate the comparison of interactions within the different *Hydra* Myc complexes. To prepare the individual systems for superposing, we restrained E-box and *Hydra* Max to allow only sidechain movements, and minimized them with the Amber10:EHT forcefield with periodic boundary conditions (AMBER 10, University of California, San Francisco, CA, USA, 2008). Structure representations were made with Schrödinger Pymol package (PyMOL Molecular Graphics System, Version 2.0, Schrödinger, LLC). Signature analysis of the *Hydra* Myc3 protein sequence was performed using the program ScanProsite (https://prosite.expasy.org/) (accessed on 20 April 2023). Additional computational protein structure analysis of the full-length *Hydra* Myc3 (A0A0H5FMB4) protein was carried out using the program AlphaFold (https://alphafold.ebi.ac.uk/) (accessed on 20 April 2023).

## 3. Results

### 3.1. Structure of the Hydra myc3 Gene and Its Protein Product

The *Hydra* Myc1 and Myc2 proteins display a highly conserved C-terminal bHLH-LZ DNA binding domain and most of the Myc-boxes in the larger N-terminal transactivation domain (Figure 1A) [18,25]. In addition to *myc1* and *myc2*, the *Hydra* genome contains the *myc3* gene, which is highly divergent. Although its predicted C-terminal bHLH-LZ domain is well conserved, the predicted N-terminal domain is short and does not contain any Myc-boxes (Figure 1A,B). Computational protein structure analysis of the full-length *Hydra* Myc3 using the program AlphaFold predicts a largely unstructured N-terminus, whereas the C-terminus clearly shows the typical signature of a bHLH-LZ domain. Protein motif analysis of the *Hydra* Myc3 N-terminus predicts the occurrence of a casein kinase II phosphorylation site (pos. 21), two protein kinase C phosphorylation sites (pos. 39/105), and one N-glycosylation site (pos. 55). Alignment of the *Hydra* Myc3 protein sequence with that of Myc1 and Myc2 showed that in all three *Hydra* paralogs the C-terminal domain encompassing dimerization surface and DNA binding regions are very similar to the human MYC (c-Myc) or viral Myc (v-Myc) proteins (Figure 1B). Phylogenic tree analysis using the C-terminal bHLH-LZ domains of transcription factors of the Myc-Max-Mlx network as well as related Mitf and Usf proteins confirmed that Myc3 is a member of the Myc protein family (Appendix A). The *Hydra myc3* gene is located on chromosome 2 and contains three exons (Appendix A), and the predicted 1420-nt mRNA (accession no. LN868213) encodes a 197-amino acid protein, which is significantly shorter than the 314-aa Myc1 or 332-aa Myc2 proteins. Similar to the regulatory sites of *myc1* and *myc2*, the putative *myc3* promoter contains TBE motifs representing binding sites for the transcription factor Tcf (TCF) (Appendix A) suggesting that *myc3* may also be regulated by canonical Wnt signaling, as was described for the *myc1* and *myc2* genes in [34]. Further inspection of the *myc3* promoter revealed the presence of binding motifs for additional transcriptional enhancers such as GATA-1 or C/EBPα (Appendix A).

### 3.2. Expression of the Hydra myc3 mRNA

Figure 2A shows a summary of the cell type-specific expression patterns of *myc3*, *myc2*, and *myc1* on a UMAP projection from the published and annotated *Hydra* single-cell RNA sequencing atlas [28]. Corresponding UMAPs for the individual genes providing more details are shown in Appendix A. *myc3* (green) is expressed in a distinct population of precursor cells committed to nerve cell and gland cell differentiation (Figure 2A,B and Appendix A). Fully differentiated nerve and gland cells do not express *myc3*. *myc2* (blue) is expressed in all proliferating cells in *Hydra*. This includes interstitial stem cells, early proliferating nematoblast nests, gland cells, proliferating precursors of both types of gametes, and proliferating epithelial cells (Figure 2A,B and Appendix A). *myc1* (red) is also expressed in interstitial stem cells, but its expression level increases in nematoblast nests, which are cells committed to differentiating into large numbers of stinging cells (nematocytes) (Figure 2A,B and Appendix A). The expression patterns of *myc1* and *myc2* shown in the single-cell RNA-seq atlas are consistent with the previously described expression patterns based on in situ hybridization experiments [18].

Whole-mount in situ hybridization experiments revealed the localization of *myc3*-positive precursor cells throughout the gastric region and not in the differentiated parts of the head and foot (Figure 2C), which is consistent with their expression in nerve and gland cell precursors. In situ images at various stages of asexual bud formation demonstrated how this spatial pattern emerges, specifically how the development of a new head and a new foot in a bud results in the disappearance of *myc3*-positive cells from these tissues (Appendix A). During nerve cell differentiation in *Hydra*, a committed interstitial stem cell becomes a nerve cell precursor that undergoes a terminal mitosis. The resulting pair of daughter cells then differentiates into mature nerve cells within 4–6 h [36]. Most of the *myc3*-expressing cells occurred as single cells and pairs, indicating that *myc3* is transcriptionally active in pre- and post-mitotic precursor stages (Figure 2C,D). At low frequency, we also observed *myc3*-expressing cell clusters containing four or eight cells, confirming earlier observations that, although rare, nerve cell precursors can undergo one or two additional rounds of replication (Figure 2C,D) [37,38].

### 3.3. Biochemical Properties of the Hydra Myc3 Protein

We have previously shown that the *Hydra* Myc1 and Myc2 proteins show the principal biochemical functions of the bilaterian Myc proteins, such as dimerization with Max and binding to double-stranded DNA [18,25]. To investigate the Myc3 C-terminus for DNA binding, the highly conserved coding sequence of the Myc3 carboxyl-terminal region (amino acid residues 108–197) encompassing the dimerization and DNA binding domain (bHLH-LZ) was inserted into the prokaryotic pET21a expression vector providing a methionine start codon and a C-terminal histidine-tag to facilitate protein purification. An analogous construct was generated using the corresponding bHLH-LZ coding region of *Hydra* Max (amino acid residues 29–120). The proteins hyMyc3-bHLH-LZ-HIS (*M*_r_ = 11,840) and hyMax-bHLH-LZ-HIS (*M*_r_ = 11,836) with apparent molecular weights of ~13,000 (p13) were efficiently expressed in *Escherichia coli*, and the soluble fractions purified in one step, using metal-chelate affinity chromatography (Appendix A). The recombinant Myc3 p13 derivative was then tested, together with the corresponding Max p13 derivative, using electrophoretic mobility shift analysis (EMSA) with increasing protein concentrations and constant amounts of DNA (Figure 3). The EMSA analysis showed that Myc3 p13 in complex with Max p13 efficiently binds to double-stranded DNA containing the authentic Myc binding site derived from the *Hydra myc2* promoter [25,34] (Figure 3), and to a DNA fragment containing the E-box of the upstream transcription factor 1 (USF) binding site [39] (Appendix A). The analysis also showed that Myc3 p13 homodimers inefficiently bind to DNA, in contrast to Max p13 homodimers (Appendix A). Because the proteins and the relevant protein–DNA complexes differ in size, it is obvious that the DNA-bound complex seen in Figure 3 must be a heterodimer between Myc3 p13 and Max p13. For quantification of the observed protein–DNA interaction, the ratios of bound to total DNA were determined and the dissociation constant (K_d_) for the protein–DNA complex was calculated. The estimated K_d_ value for the protein–DNA complex formed by Myc3 p13/Max p13 was determined to be 1.63 × 10^−8^ M, which is in the range of the previously determined dissociation constants of the *Hydra* Myc1/Max (1.70 × 10^−8^ M) and Myc2/Max (1.74 × 10^−8^ M) heterodimers [18,25]. This indicates that the *Hydra* Myc3 p13/Max p13 dimer binds with a similar or slightly higher affinity to specific DNA, and that in vivo expressed Myc3 could efficiently compete with Myc1 or Myc2 for heterodimerization with Max.

### 3.4. Oncogenic Potential of the Hydra Myc3 Protein

Recently, we have reported that hybrid proteins between *Hydra* Myc1 or Myc2, and viral Myc (v-Myc) have cell transforming potential [18,25]. To explore whether the *Hydra* Myc3 protein also displays some of the principal biological functions of vertebrate Myc, the *myc3* coding region and hybrids between *myc3* and v-*myc* were inserted into the replication-competent retroviral RCAS vector. In these constructs, the *myc3* coding sequences of the N-terminal region and the C-terminal DNA binding domain were mutually exchanged (hymyc3/v-myc, v-myc/hymyc3), and compared with two analogous v-*myc/myc1 and* v-*myc/myc2* hybrids [25] for their capacity to induce cell transformation in avian fibroblasts. The empty RCAS vector and the RCAS-v-myc construct encoding the 416-amino acid viral Myc (v-Myc) protein were used as controls, together with an RCAS construct encoding the full-length *Hydra* Myc3 protein supplied with an N-terminally attached HA-tag (Figure 4A). The retroviral constructs were transfected into primary quail embryo fibroblasts (QEF), and the cells were passaged several times.

To test for their transformed phenotypes, cells were then seeded into soft agar, and colony formation was monitored after two weeks (Figure 4A). Unexpectedly, cells expressing the hybrid v-Myc/hyMyc3 protein displayed a strong transforming potential, comparable to that of the original v-Myc oncoprotein, whereas the corresponding v-Myc/hyMyc2 or v-Myc/hyMyc1 proteins induced partial cell transformation manifested by lower agar colony numbers, in accordance with previous results [25]. Likewise, cells transformed by the v-Myc/hyMyc3 chimera showed a similar morphology and a high proliferation rate, comparable to v-Myc-transformed cells (Figure 4A and Appendix A) and were able to induce focus formation with an efficiency comparable to v-Myc (Appendix A). Neither significant morphological alterations nor a transformed phenotype was monitored in cells transfected with the empty RCAS vector and in cells expressing the full-length HA-tagged hyMyc3 protein, or in cells expressing chimeric hyMyc3/v-Myc (Figure 4A and Appendix A). The latter results were expected, due to the lack of a Myc-homologous N-terminal domain in the *Hydra* Myc3 protein (cf. Figure 1). Expression of the ectopic viral and hybrid Myc proteins was monitored by immunoblot analysis, using antibodies specific for the N-terminal or C-terminal v-Myc regions, and an antibody directed against the N-terminal HA-tag of the Myc3 fusion protein (Figure 4B). Expression levels of v-Myc/hyMyc2 and v-Myc/hyMyc1 were slightly lower than those of v-Myc/Myc3 or v-Myc, but previous results had shown that even equal protein expression levels did not result in higher cell transforming capacities of v-Myc/myc1 or v-Myc/Myc2 compared to v-Myc [25]. Altogether, these results show that, in contrast to *Hydra* Myc1 or Myc2, the C-terminal domain of *Hydra* Myc3 has about the same transformation capability as the highly oncogenic retroviral v-Myc protein.

### 3.5. Homology Modeling of Hydra Myc3 Predicts a High Stability of Myc3/Max Heterodimers

In order to obtain hints about the higher transforming potential of the Myc3 C-terminus compared to Myc2 or Myc1 on a structural basis, we performed homology modeling of this domain, using as a template the 3D structure of the corresponding region of the human MYC/MAX heterodimer in complex with a MYC binding site [35]. The analysis revealed specific conservation of residues at the contact surfaces and led to the identification of amino acid residues contributing to heterodimer stabilization of the different hyMyc variants, thereby predicting that the *Hydra* Myc3/Max complex may form the most stable dimer (Figure 5). Homology modeling followed by energy and geometry optimization showed structural differences in the relevant binding interactions. *Hydra* Myc proteins bind to DNA mainly through polar interactions, whereas Max is stabilized mostly by hydrophobic contacts. The surface of the *Hydra* Myc3/Max complex shows that the lower DNA binding part is more polar than the upper polypeptide-associating region (Figure 5A). The N-terminal parts of the modeled peptides contain more positive charges near the DNA framework to attract phosphoric acid groups and to establish polar contacts with nucleotide base pairs and surrounding water molecules. On the other hand, the Max binding site comprises more hydrophobic residues, even though a few polar contacts remain essential (Figure 5B). When comparing molecular interactions within the *Hydra* Myc family, superposition of the *Hydra* Myc variants suggests that *Hydra* Myc3 may bind to *Hydra* Max with the highest efficiency (Figure 5C). In particular, electrostatic repulsions within the DNA framework and mismatches of polar/hydrophobic residues render Myc1 and Myc2 less favorable for heterodimer formation, typically at the following sites. In Myc3 there are stabilizing hydrogen bonds between residue N126 and phosphoric acid oxygen atoms from the DNA, in contrast to Myc2 or Myc1, which contain at this position an I263 or D238, respectively (Figure 6). Whereas Myc3 and Myc2 possibly form stabilizing salt bridges between residues K155 or K292, respectively, and D48 from *Hydra* Max, the corresponding T267 from Myc1 cannot form this type of interaction. Furthermore, the Myc3 L175 attracts I92 in *Hydra* Max via hydrophobic interactions, which is not possible for the corresponding K312 or K287 in Myc2 or Myc1, respectively (Figure 6).

Altogether, the results from the structural modeling predict that the stabilities of *Hydra* Myc/Max heterodimers may increase by the order Myc1 < Myc2 < Myc3, although the gel-based EMSA analyses could only detect marginal differences in the dissociation constants of Myc1 [18], Myc2 [25], or Myc3 (cf. Figure 3).

### 3.6. Conservation of Critical Amino Acid Residues in the Hydra Myc3 Leucine Zipper Region

The strikingly high transforming potential of the *Hydra* Myc3 C-terminus compared to the corresponding Myc2 and Myc1 regions prompted us to identify critical amino acid residues in the dimerization and DNA binding region. Alignment of the relevant *Hydra*, viral, and human(hu) Myc polypeptide sequences revealed that the bHLH-LZ region of Myc3 shares a 52%/51% sequence identity with the homologous v-Myc/huMYC region, whereas the corresponding sequence identities between Myc2 and v-Myc/huMYC, or Myc1 and v-Myc/huMYC, account for 41%/49% and 40%/47%, respectively (cf. Figure 1B). Therefore, the *Hydra* Myc3 C-terminal region offers the closest homology to its viral or human counterparts. Concerning the dimerization surface, four amino acid residues in the human MYC leucine zipper (LZ) region (E410, E417, R423, R424) confer steric and electrostatic repulsions, which are responsible for inefficient MYC homodimerization but favor the formation of MYC/MAX heterodimers [40]. These findings have led to the development of a competitive dominant negative MYC inhibitor termed Omomyc, which was applied to treat MYC-dependent tumor cells [41]. Interestingly, comparison of orthologous *Hydra* Myc protein regions with those from human or viral Myc showed that three of the four residues are conserved in *Hydra* Myc3 (E410, R423, R424), whereas Myc2 (E410) and Myc1 (E410D) each contain only one conserved residue (Appendix A).

To test if mutations at these residues affect the transforming potential of Myc in our cell system, mutagenesis was performed using the retroviral expression plasmid pRCAS(A)BP-HA-MYC as a template encoding an N-terminally HA-tagged human MYC protein [31]. The resulting expression plasmids pRCAS(A)BP-HA-c-myc-QN and pRCAS(A)BP-HA-c-myc-TIQN encode human HA-MYC proteins with the mutations R423Q/R424N (QN) and E410T/E417I/R423Q/R424N (TIQN) and thus carry the same mutations found in the Omomyc inhibitor (Figure 7A). The constructs were transfected into QEF, and the resulting cell cultures were tested for their transformed phenotype using focus and agar colony formation. Immunoblot analysis from parallel cultures showed efficient expression of all HA-tagged ectopic MYC proteins (Figure 7B).

After transfection, cells were overlayed with agar, and then stained after two weeks to visualize the emergence of cell foci (Figure 7C). A significant reduction in focus formation was observed in the case of the TIQN mutant, whereas substitution of only two residues (QN) was not sufficient to strongly interfere with the oncogenic potential of MYC. Moreover, when parallel cell cultures were passaged several times, the morphology of cells expressing HA-MYC(QN) or HA-MYC(TIQN) was more similar to cells expressing wild-type HA-MYC than to untransformed cells transfected with the empty RCAS vector (Figure 7C). Testing for agar colony formation confirmed that substitution of these dimerization-sensitive residues in human MYC does not completely block cell transformation, but rather induces a partially transformed phenotype. However, the result suggests that the higher conservation grade of these four critical residues in *Hydra* Myc3 contributes to the intrinsically high transforming potential of the Myc3 C-terminus upon fusion with an authentic Myc N-terminal domain (cf. Figure 4 and Appendix A).

## 4. Discussion

Myc is a cellular master regulator with pleiotropic functions, which has been intensively studied in recent decades to elucidate many aspects of human MYC biology. However, there still remain multiple open questions with respect to the molecular mechanisms by which Myc controls cellular proliferation, growth, metabolism, programmed cell death, or differentiation [5,11,42,43,44]. Dissecting the multiple roles of Myc benefits from the use of defined genetic systems in which conserved functions can be addressed. Like many other oncogenes, *myc* was originally isolated from a transforming chicken retrovirus. This viral *myc* allele, termed v-*myc*, was transduced from the cellular gene c-*myc*, which was later identified virtually by its homology to viral *myc* [3,5,6]. Therefore, using avian cell systems derived from chicken or quail embryo fibroblasts is highly suitable for functional analyses of Myc’s oncogenicity. In these cells, activated *myc* alleles are sufficient to induce neoplastic transformation and tumorigenesis within days. This is in contrast to primary mammalian cells, which require the presence of a second cooperative oncogene to achieve neoplastic transformation [45]. Here, we combine the use of avian cell lines with structural variants of the *myc* proto-oncogene naturally occurring in a cnidarian to reveal insights into the relationship between the Myc protein structure and function.

Cnidarians such as *Hydr*a are the sister group to bilaterians, which branched off before the Cambrian radiation more than 600 M years ago. Cnidarians are relatively simple animals, with a small number of cell types and simple body plans. *Hydra* is commonly used to investigate pattern formation, regeneration, and stem cell dynamics [21,23,24,46,47,48] and offers powerful molecular tools and resources [20,49,50,51]. More recently, it has also been used to study the cancer driver Myc and its upstream-acting signaling network, thereby demonstrating that principal biochemical and oncogenic functions of Myc arose very early in metazoan evolution [18,25,52]. The *Hydra* genome encodes four *myc*-related genes (*myc1-4*). *myc1* and *myc2* show the closest homology to vertebrate c-*myc*, while *myc3* and *myc4* are significantly less conserved [18,20,25]. Whereas Myc1 and Myc2 proteins display the same principal topography as compared with the human MYC protein [18], full-length Myc3 displays a shortened N-terminal transactivation domain, and lacks all the conserved Myc boxes (cf. Figure 1). While it is likely that the earliest metazoans had only one *c-myc* gene, *myc* genes have diversified in all cnidarian lineages, resulting in between four and seven paralogs being present in most extant species. Moreover, loss of N-terminal Myc boxes occurred in *myc* paralogs in all major cnidarian lineages, and we detected *myc* genes of the Myc3-type completely lacking N-terminal Myc boxes in all classes of this phylum. A study describing the detailed evolutionary history and dynamics of the Myc protein family in cnidaria and across all metazoa is in progress. 

### 4.1. Myc/Max Dimerization and Oncogenic Transformation

The presence of a *myc*-related gene in *Hydra* encoding a Myc protein with fully functional dimerization and DNA binding domain but a truncated transcriptional regulation domain increases the complexity of the known Myc/Max dimerization network. The structural and functional analysis of *Hydra myc3* presented here revealed that the encoded Myc3 protein efficiently dimerizes with *Hydra* Max and binds to a canonical Myc binding site. Surprisingly, the C-terminal dimerization and DNA binding domain of Myc3 offers the highest structural conservation in comparison to Myc1 or Myc2. This may be the reason why Myc3 derivatives containing the v-Myc N-terminus display a transforming potential comparable to that of vertebrate Myc orthologues.

Combining the data from previous [18,25] and present (cf. Figure 3) protein–DNA interaction studies, we observed that all *Hydra* Myc/Max heterodimers bind with comparable efficiencies to DNA with K_d_ values in the range of 1–2 × 10^−8^ M. However, the results from the molecular modeling analysis (cf. Figure 5 and Figure 6) suggest that Myc3/Max heterodimers have a higher stability compared to Myc2/Max or Myc1/Max dimers. A possible reason for this discrepancy might be that the gel-based EMSA technique only allows estimation, but not precise determination, of dissociation constants from protein–DNA complexes for the following reasons. During gel electrophoresis, samples are no longer in chemical equilibrium where rapid dissociation can prevent complex detection, and slow dissociation can result in underestimation of the binding density [53]. On the other hand, many complexes are significantly more stable in the gel matrix as they are in free solution, which limits studies to measure reaction kinetics with larger relaxation times [53]. If the individual *Hydra* Myc proteins vary in these parameters, as suggested by structural modeling (cf. Figure 5 and Figure 6), the differences are not recorded by using the applied classical protein–DNA detection technique. Gel-free alternatives for quantification of protein–DNA interactions have been developed, such as surface plasmon resonance platforms or microscale thermophoresis, as demonstrated recently using recombinant MYC and MAX proteins [54]. These biophysical techniques would represent an option for the future to precisely quantify *Hydra* Myc/Max dimerization and DNA interaction in free solution.

The high conservation grade of *Hydra* Myc3 on the leucine zipper dimerization surface prompted us to analyze the functional relevance of specific residues, thereby showing that those are required for full cell transformation (cf. Figure 7). These residues have been originally identified as crucial for heterodimerization with Max, and were mutated to create a dominant negative Myc polypeptide. This molecule, termed Omomyc, encompasses the bHLH-LZ region of MYC with four amino acid substitutions conferring different dimerization properties. Omomyc interferes with Myc/Max dimerization and DNA binding by competitive inhibition and sequestration of oncogenic Myc [55,56], and overexpression of Omomyc inhibits Myc-mediated transcription and cell transformation [40]. Likewise, the application of Omomyc as a cell-penetrating peptide in patients suffering from solid cancers has provided promising results in a first phase-I clinical trial [57]. The high conservation of these charged residues in the leucine zipper region of Myc3 compared to Myc1 or Myc2 suggests that the *Hydra* Myc proteins have different dimerization properties (cf. Figure 5 and Figure 6), which could represent a structural reason for the stronger intrinsic transforming potential of Myc3, compared to Myc2 or Myc1 (Figure 8A).

### 4.2. A Possible Role for Hydra Myc3 in Balancing Stemness and Differentiation

*myc3* is specifically expressed in progenitor cells committed to nerve and gland cell differentiation, while *myc2* expression is downregulated in these cells; *myc2* is associated with cell cycle progression and may mediate stemness in interstitial stem cells. Another transcription factor-encoding gene, *myb*-like (G020130) (formerly *myb*), exhibits a *myc3*-equivalent expression pattern in nerve and gland cell precursors [29]. In terminally differentiated nerve and gland cells, *myc3* and *myb*-like expression is down-regulated. These data suggest that *myc3* is part of a transcriptional program directing interstitial stem cells toward the neuron- and gland-cell fates. One of the upstream players activating *myc3* in the precursor cells could be β-Catenin/Tcf signaling. Several findings support this view: (1) pharmacological activation of β-Catenin activates neuronal marker genes in *Hydra* [58]; (2) neuron density in the tissue in β-Catenin over-activated β-Cat-Tg transgenic polyps is about twice as high (own unpublished data), and (3) putative Tcf-binding elements in the *myc3* promoter are located close to the transcriptional start site (Appendix A).

Based on the data presented here, we propose a model in which Myc3/Max dimerization contributes to changes in the stemness–differentiation balance in interstitial stem cells exiting to nerve- and gland-cell differentiation (Figure 8B). Equal, or even higher, affinity for Max dimerization and DNA binding of Myc3 as compared with Myc2 indicates that effective competition for E-box Myc binding sites could take place as soon as the Myc3 protein becomes prevalent. Thereby, the equilibrium of Myc/Max heterodimers would shift away from Myc2/Max in favor of Myc3/Max, leading to a transcriptionally inactive complex, due to the lack of a functional N-terminal transactivation domain in Myc3. Consequently, the transcriptional network relevant for stem cell maintenance changes towards a network stabilizing the committed state and driving cells more towards nerve-and gland-cell differentiation (Figure 8B). At present, this model is only supported by descriptive in vivo data and in vitro biochemical data. To further test this model in the future, gene-specific gain- and loss-of-function assays should be conducted in vivo.

This strategy of using a truncated Myc protein to direct a specific interstitial differentiation pathway may be more broadly used in hydrozoans. A comparison with the single cell atlas from the hydromedusa *Clytia hemisphaerica* [30] revealed a *myc* paralog (XLOC_007085) encoding a Myc protein with reduced N-terminal Myc boxes and a highly conserved bHLH-LZ region (65% identity with human MYC), which is specifically expressed in nerve cell precursors (Appendix A). Another *Clytia myc* paralog (XLOC_000985) encodes the prototypic Myc protein with Myc boxes but a less-conserved bHLH-LZ region (56% identity with human MYC). This gene is expressed in interstitial stem cells and in a variety of other cell types, including germ cells similar to *Hydra myc2* (Appendix A). Therefore, a common ancestor of the *Hydra* and *Clytia* lineages living about 400 M years ago may have already used an N-terminally reduced Myc variant acting as a competitor for Max and DNA binding and thereby shifting the stemness–differentiation balance in interstitial stem cells towards a commitment to neuronal differentiation.

## 5. Conclusions

The characterization of a third *myc* gene in *Hydra* with conserved biochemical and oncogenic functions revealed a high complexity of the Myc network in this cnidarian. Based on its specific expression pattern, *myc3* appears to function in nerve- and gland-cell differentiation, where its protein product could act as a dominant negative competitor of the stem cell-specific Myc1 and Myc2 proteins. In contrast to its highly divergent N-terminus, the Myc3 C-terminus displays the best conservation grade compared to vertebrate Myc proteins, which may explain its strong intrinsic oncogenicity. Comparison of different Myc isoforms in terms of structure and function could therefore lead to the identification of potentially druggable surfaces on Myc, which represents an oncogenic driver in most human tumors.

## Figures and Tables

**Figure 1 cells-12-01265-f001:**
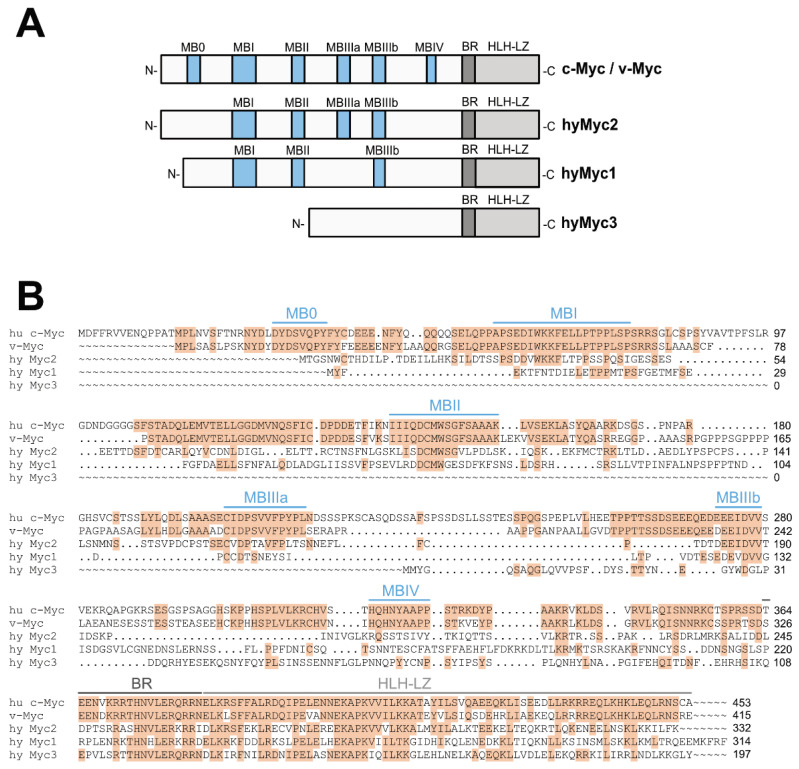
Structural relationship of human and *Hydra* Myc proteins. (**A**) Schematic depiction of the human c-Myc (MYC)/v-Myc and *Hydra* Myc1, Myc2, and Myc3 proteins. The positions of the C-terminal basic region (BR), helix-loop-helix/leucine zipper domain (HLH-LZ), and N-terminal Myc boxes (MBs) are represented in grey or blue, respectively. (**B**) Alignment of human (hu), viral (v), and *Hydra* (hy) Myc proteins using the program Clustal Omega (GenBank accession nos.: hu c-Myc, NP002458; v-Myc, P01110; hyMyc2, ADA57607; hyMyc1, ACX32068; hyMyc3, CRX73227) with shading based on the similarity to human c-Myc (MYC). Identical amino acid residues are shaded in light brown, and gaps are indicated by points. The positions of conserved Myc boxes (MBs), basic region (BR) and helix-loop-helix/leucine zipper domain (HLH-LZ) are indicated above the alignment in blue or grey, respectively.

**Figure 2 cells-12-01265-f002:**
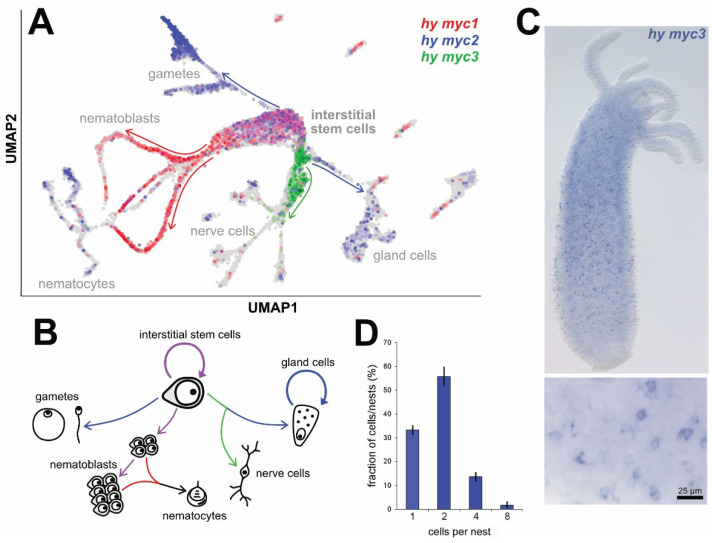
Expression patterns of the different *myc* genes in *Hydra*. (**A**) The interstitial stem cell lineage in *Hydra* resolved by single cell RNA sequencing [28,29] depicted in two-dimensional space (UMAP, uniform manifold approximation and projection). Arrows indicate differentiation paths from the interstitial stem cell pool to the differentiated cell types. Each cell is colored according to its expression (*myc1*, red; *myc2*, blue; *myc3*, green). Cells expressing none of the three *myc* transcripts are colored in grey. (**B**) Schematic representation of the multipotent interstitial stem cell system and its derivatives. Colors of the arrows correspond to the expression of the different *myc* genes at the different stages. *myc1* (red) is expressed during nematocyte differentiation, *myc2* (blue) contributes to the production of gametes and is expressed in mature gland cells, *myc3* (green) is activated in nerve and gland cell progenitors, and both *myc1* and *myc2* (purple) are involved in interstitial stem cell maintenance and early nematoblast stages. (**C**) Whole *Hydra* polyp in situ hybridization showing *myc3* expression throughout the gastric region and a magnified view of nerve and gland precursor cells occurring as single cells, as cell pairs, and, rarely, as small nests of cells. (**D**) Quantification of cell cluster size of *myc3* expressing cells. Bars represent mean ± SD from 3 polyps; >100 cells and cell clusters were counted per polyp.

**Figure 3 cells-12-01265-f003:**
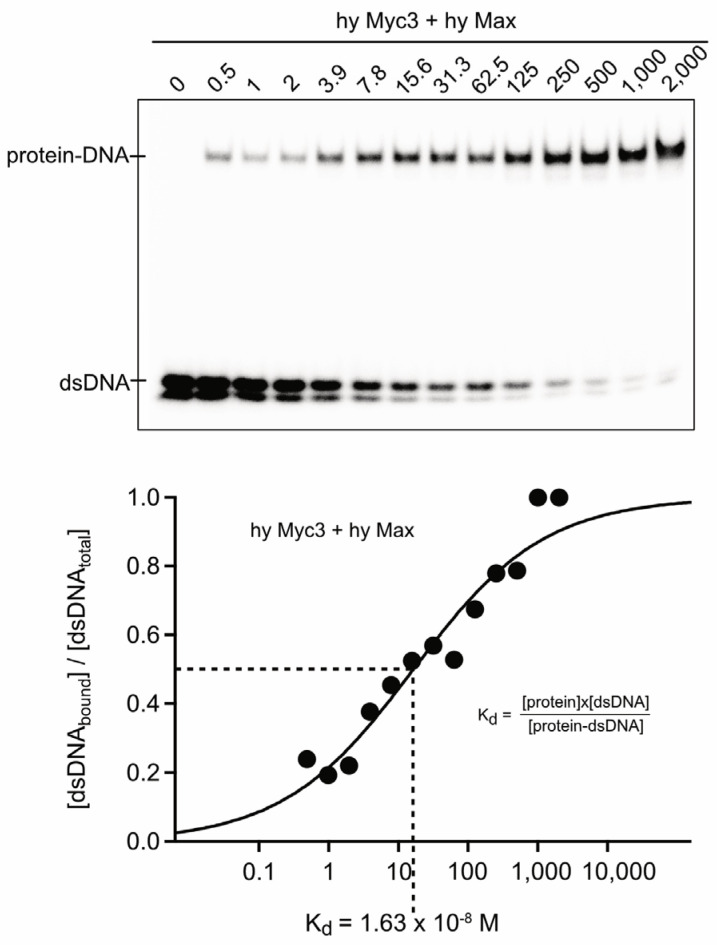
DNA binding of *Hydra* Myc3/Max dimers. Upper panel: electrophoretic mobility shift assay (EMSA) using increasing amounts (0–2000 nM) of recombinant polypeptides (p13) encompassing the *Hydra* Myc3 and Max bHLH-LZ regions, and 0.3-ng (25,000 cpm) aliquots of a ^32^P-labeled double-stranded 18-mer oligodeoxynucleotide containing the canonical Myc/Max-binding motif 5′-CACGTG-3′ (E-box) in the context of the *Hydra myc2* promoter [34]. Final protein concentrations [nM] are indicated above each lane. Lower panel: determination of the dissociation constants (K_d_) of the protein–DNA complexes after titration with increasing amounts of proteins. The ratios of bound DNA to total DNA were determined by phosphor-imaging, and plotted versus the log_10_ of the applied protein concentrations. Because the experimental conditions led to partial DNA strand separation, only double-stranded DNA was considered for the quantification of unbound DNA. The sigmoidal fit function f(x) = 1/{1 + exp[(a − x)/b]} [25] was used to generate the binding curve. The calculated K_d_ value for the protein–DNA binding reaction is indicated below.

**Figure 4 cells-12-01265-f004:**
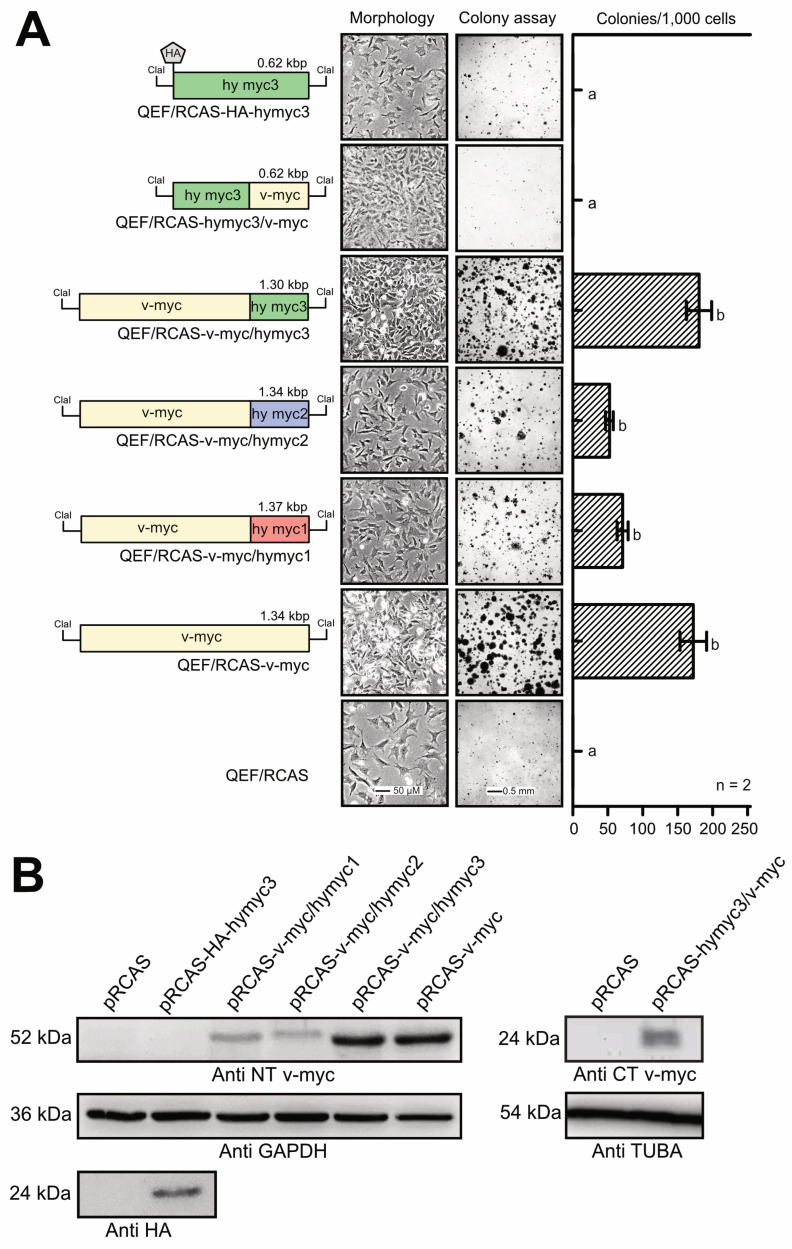
Cell transforming capacity of *Hydra* Myc3. (**A**) Left site: schematic depiction of the coding sequences from HA-tagged full-length hymyc3, hy/v-myc and v/hy-myc hybrids, and v-myc (green, hymyc3; blue, hymyc2; red, hymyc1; yellow, v-myc). The coding regions were inserted into the unique *Cla*I site of the replication-competent retroviral pRCAS vector used for DNA transfection into primary quail embryo fibroblasts (QEF). Right site: cell morphology and transformed phenotype of the transfected and passaged QEFs. Phase-contrast micrographs to monitor cell morphology (left panel). Colony assay to monitor cell transformation (right panel). Each aliquot of 1.25 × 10^4^ cells were seeded into soft agar on MP24 wells and incubated for two weeks. Numbers of colonies per 1000 seeded cells are shown in the column graph. The numbers of colonies were compared using an unpaired *t*-test (n = 2), where *a* and *b* are significantly different (*p* < 0.05). (**B**) Immunoblot analysis to monitor ectopic Myc protein expression using extracts from the cells shown in (**A**). Antibodies are directed against the N-terminal or C-terminal part of v-Myc to detect the different hybrid proteins. Antibodies directed against GAPDH or α-tubulin (TUBA) were used as loading controls. Proteins were resolved by SDS-PAGE (10%, wt/vol).

**Figure 5 cells-12-01265-f005:**
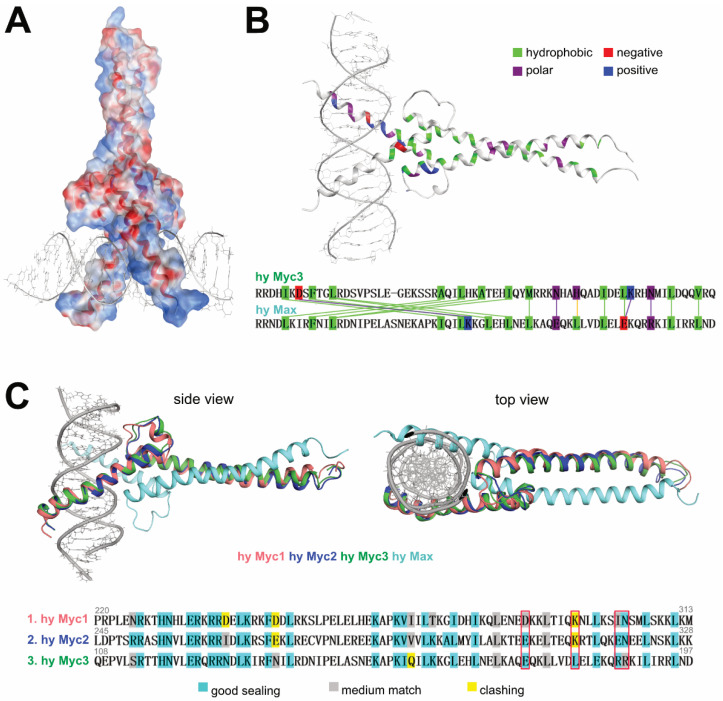
Template-based structure model of the bHLH-LZ regions of *Hydra* Myc and Max proteins and interaction analyses of *Hydra* Myc/Max binding interfaces. (**A**) Homology model of the *Hydra* Max/Myc3/E-box binding complex based on the human Myc/Max/DNA structure (PDB accession code 1NKP). Surface representation with colors based on charges (blue, positive; red, negative). A high number of negative charges is present at the binding interface with the DNA, favorizing the stabilizing salt bridges of the phosphoric acid moieties present within the DNA/E-box frame. The dimerization surface of Myc3 contains fewer charges and interacts with Max mainly by hydrophobic interactions. (**B**) Residue-wise analysis of *Hydra* Myc3/Max/E-box binding interactions. Residues are colored based on their physico-chemical properties (green, hydrophobic residues; blue, positive charges: red, negative charges; purple, neutral polar residues). Polar residues of Myc3 are more involved in DNA binding, whereas hydrophobic residues dominate in the Max interaction surface. (**C**) Superposition of *Hydra* Myc1, Myc2, and Myc3 in complex with Max binding to DNA. Within the *Hydra* Myc protein family, the DNA binding quality of the different Myc/Max complexes can be distinguished by monitoring electrostatic clashes. The efficiency of Max dimerization depends on both the hydrophobic and electrostatic complementarities of the relevant residue matches. According to this model structure, the Myc3/Max dimers are more stable than the Myc1/Max or Myc2/Max dimers, due to fewer electrostatic repulsions, which are visualized in the alignment below. Corresponding residues in human MYC, which were used for further mutational analysis, are indicated by red boxes.

**Figure 6 cells-12-01265-f006:**
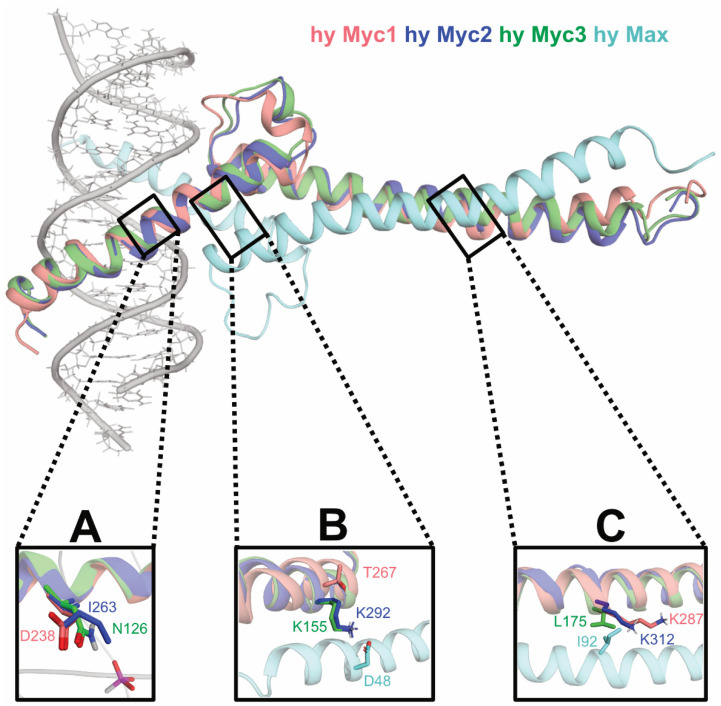
Exemplary areas with varying E-box/Max interaction profiles within the *Hydra* Myc family. (**A**) Myc3 forms stabilizing hydrogen bonds between residue N126 and the phosphoric acid groups present in the DNA framework. In contrast, the corresponding I263 residue in Myc2 does not further stabilize the binding interface, and the corresponding D238 in Myc1 even repulses negatively charged oxygen atoms from the phosphodiester linkage. (**B**) Myc3 and Myc2 form stabilizing salt bridges between the residues K155 or K292, respectively, and D48 of *Hydra* Max, whereas T267 of Myc1 cannot form this stabilizing interaction. (**C**) The Myc3 L175 attracts the I92 of Max via hydrophobic interactions, in contrast to Myc2 and Myc1, which contain at this position the hydrophilic residues K312 or K287, respectively.

**Figure 7 cells-12-01265-f007:**
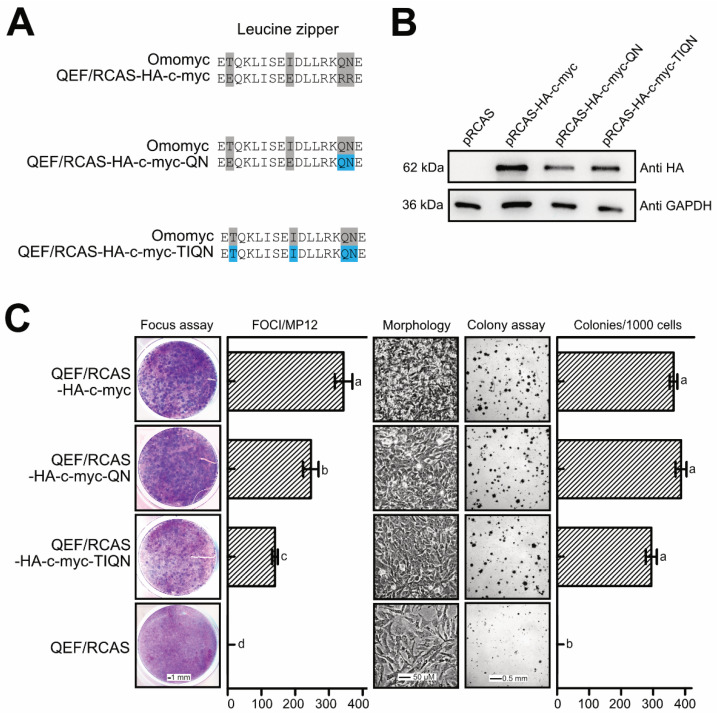
Cell transformation interference caused by leucine zipper-mutations in human MYC. (**A**) Amino acid sequences of the leucine zipper regions in which Omomyc-specific mutations [40] (shaded in grey) have been introduced by site-directed mutagenesis into the *MYC* coding sequence in the construct pRCAS-HA-c-myc (shaded in blue). (**B**) HA-MYC protein expression monitored by immunoblot analysis using extracts from QEF transfected with RCAS constructs encoding HA-tagged human MYC wild-type and mutant constructs after several passages. Antibodies specific for the HA-tag or GAPDH were applied. Proteins were resolved by SDS-PAGE (10%, wt/vol). (**C**) Cell transformation assays of QEF transfected with the same RCAS constructs as above. Each 1 µg of plasmid DNA was transfected into QEF grown onto MP12 wells and then kept under agar overlay for 2 weeks, followed by staining with eosin methylene blue (left panel). The numbers of foci were compared using an unpaired *t*-test (n = 2), where *a*, *b*, *c* and *d* are significantly different to each other (*p* < 0.05). For mass cultures, each 4 µg of plasmid DNA was transfected into QEF grown on 60 mm dishes. Cells were passaged several times and phase-contrast micrographs were taken to visualize cell morphologies (right panel). Equal numbers of these cells (5 × 10^3^) were seeded into soft agar on MP24 wells and incubated for 2 weeks. Numbers of colonies per 1,000 cells seeded are shown next to the bright-field micrographs. The numbers of colonies were compared using an unpaired *t*-test (n = 2), where *a* and *b* are significantly different (*p* < 0.05).

**Figure 8 cells-12-01265-f008:**
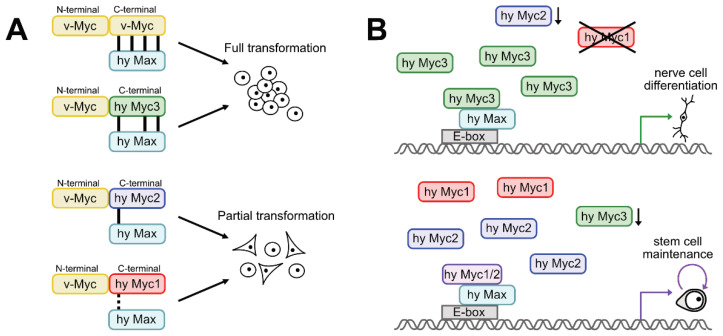
Models comparing the actions of the three *Hydra* Myc proteins in cell transformation and differentiation. (**A**) Transforming activities of the different protein chimera compared to the original v-Myc protein. Chimeric v-Myc/hyMyc3 induces full cell transformation, contrary to v-Myc/hyMyc2 or v-Myc/hyMyc1, which only partially transform cells. Bars between different heterodimers depict the conservation grade of amino acid residues relevant for Myc’s heterodimerization potential (cf. Figure 5 and Figure 6) and for its oncogenic capacity (cf. Figure 4). Each bar represents one of four Myc residues required for Myc/Max interaction (from left to right in hyMyc3: E410, R423, and R424), which are identical for the relevant *Hydra* Myc protein and v-Myc. The dotted bar indicates chemical conservation between two residues (E/D). The fourth residue, E417, is not conserved between v-Myc and human MYC (not shown). (**B**) Proposed model of how the *Hydra* Myc proteins regulate proliferation and differentiation in *Hydra*. Upper panel: Myc3/Max dimers dominate binding to E-box signatures, due to transcriptional down-regulation of *myc2* and *myc1* in committed nerve and gland cell precursors and high affinity of Myc3 for interaction with Max. This induces a shift in gene expression, leading to nerve and gland cell differentiation. Lower panel: Myc3 is transcriptionally repressed in the stem cell pool, and Myc2/Max and Myc1/Max bind to the promoters of relevant target genes and thereby mediate stem cell proliferation and maintenance.

## Data Availability

All data generated in this study are available on request.

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
