# Peer review of "High Intrinsic Oncogenic Potential in the Myc-Box-Deficient Hydra Myc3 Protein"

_cells, 2023, doi:10.3390/cells12091265_

Round 1

Reviewer 1 Report

Lechable et al. Myc in Hydra

This is a well-done paper examine the function of a form of Myc, Myc3, in hydra. The primary finding is that Myc 3 lacks the N-terminal conserved Myc-boxes classically required for cell transformation, but the C-terminal BHLHZip domain is conserved. Consequently, is perhaps not surprising that Myc3 cannot transform quail cell unless the N-terminal domain is replaced with that of V-Myc. I am not familiar with the genetic tricks that can be employed in hydra, so I can’t really comment on potential in vivo experiments that would add to the manuscript. But, the data presented are convincing that Myc3 can interact with Max and localize with transformation relevant targets in avian cells. I am not particularly convinced that Myc1 and Myc2 are fundamentally different from Myc3 in terms of their dimerization and DNA-binding capacity ( and consequently are slightly less effective in transformation assays) when complexed with Max as assayed by modeling approaches and in an orthogonal transformation assay. Those critiques aside, I found the paper interesting to read, well written and believe that it makes a valuable contribution to the literature.

Major points

1.     Classically, the inclusion of the N-terminal Myc-boxes defines a Myc paralog. Myc3 lacks these, so can Myc3 really be called a Myc family protein?.  How do other CACGTC-binding BHLHZip proteins, e.g. USF, TFE3, compare to the BHLHZip domain of Myc3. Could Myc3 just be a paralog of one these other BHLHZip CACGTG binders? I believe that some of these proteins can even interact with Max under the right protein concentrations.

2.     Without functional testing, the modeling experiments that predict that Myc3:Max complex are more stable than Myc1:Max and Myc2:Max complexes are not compelling. Especially, when the authors can;t detect differences in Kd using quantitative gel shift assays.  I don’t think functional testing has to be done in this paper, but perhaps soften the language around this point.  One could imagine looking at relative abundance of the different Myc:Max complexes by coIP or measuring their on-and-off rates at CACGTG sites in a gel shift format.

Author Response

This is a well-done paper examine the function of a form of Myc, Myc3, in hydra. The primary finding is that Myc 3 lacks the N-terminal conserved Myc-boxes classically required for cell transformation, but the C-terminal BHLHZip domain is conserved. Consequently, is perhaps not surprising that Myc3 cannot transform quail cell unless the N-terminal domain is replaced with that of V-Myc. I am not familiar with the genetic tricks that can be employed in hydra, so I can’t really comment on potential in vivo experiments that would add to the manuscript. But, the data presented are convincing that Myc3 can interact with Max and localize with transformation relevant targets in avian cells.

We thank the referee for this constructive review. Below are our specific responses to the comments.

I am not particularly convinced that Myc1 and Myc2 are fundamentally different from Myc3 in terms of their dimerization and DNA-binding capacity ( and consequently are slightly less effective in transformation assays) when complexed with Max as assayed by modeling approaches and in an orthogonal transformation assay.

We agree with this concern, although we observed a quantitative higher transforming potential of the v-Myc/Myc3 chimera compared to v-Myc/Myc2 or v-Myc/Myc1. Also, the morphology of v-Myc/Myc3-transformed fibroblast resembles more to those transformed by the original v-myc oncogene (cf. Figure 3A). We admit that the differences in DNA binding of the different Myc isoforms may not be large, but Myc3/Max could still efficiently compete with DNA binding of Myc2/Max or Myc1/Max heterodimers.

Those critiques aside, I found the paper interesting to read, well written and believe that it makes a valuable contribution to the literature.

Major points

  1. Classically, the inclusion of the N-terminal Myc-boxes defines a Myc paralog. Myc3 lacks these, so can Myc3 really be called a Myc family protein?.  How do other CACGTC-binding BHLHZip proteins, e.g. USF, TFE3, compare to the BHLHZip domain of Myc3. Could Myc3 just be a paralog of one these other BHLHZip CACGTG binders? I believe that some of these proteins can even interact with Max under the right protein concentrations.

When preparing this manuscript, we thought about this issue, and about how to name this unique Hydra Myc protein. Three lines of evidence support the view, that this is a bona fide, but derived, Myc factor: (1) molecular phylogenetic analysis, (2) the presence of amino acid residues in the bHLH-LZ domain diagnostic for Myc proteins (according to Atchley and Fernandes, 2005, PNAS 102, 6401-6406), and (3) a broader Myc diversification in the cnidarian phylum, that we mention in the discussion, and that resulted in an evolution of Myc box-deficient Myc proteins in several cnidarian classes. A phylogenetic tree of the bHLH-LZ regions from Hydra Myc isoforms and from other bHLH-LZ Hydra proteins was now added (new Figure A1) showing that Myc3 displays the closest relationship to Myc proteins and significantly less homology to other bHLH-LZ proteins such as for instance Hydra MITF. Therefore, we conclude that Myc3 clearly belongs to the Myc family.  We cannot exclude that Myc3 could interact as well with other bHLH-LZ proteins not related to Myc and Max. However, the Hydra single cell atlas shows that myc3 and max, different to other bHLH-LZ factor encoding genes, exhibit specific and high levels of gene expression in this defined population of committed nerve and gland precursor cells.

  1. Without functional testing, the modeling experiments that predict that Myc3:Max complex are more stable than Myc1:Max and Myc2:Max complexes are not compelling. Especially, when the authors can;t detect differences in Kd using quantitative gel shift assays.  I don’t think functional testing has to be done in this paper, but perhaps soften the language around this point.  One could imagine looking at relative abundance of the different Myc:Max complexes by coIP or measuring their on-and-off rates at CACGTG sites in a gel shift format.

We have modified the description of the structural modeling results (Figures 5, 6) accordingly thereby avoiding strong statements and emphasize in the discussion that these results still have to be verified by appropriate biochemical experiments.

Reviewer 2 Report

This is an interesting study that examines the role of one of 4 Myc proteins from the freshwater cnidarian polyp Hydra, namely Myc 3.  Myc 1 and Myc 2 are closely related the c-Myc of vertebrate, whereas Myc 3 is highly divergent and lacks most of the N-terminal transactivation domain of other Myc proteins and v-myc. Moreover, either because of the lack of the TAD or evolutionary divergence, it contains none of  “Myc boxes” that are found in other Myc proteins. Myc3 does dimerize strongl with Max and can bind E boxes in vitro. The Myc3 expression pattern in Hydra suggests a distinct role in cell differentiation that may involve Myc3 competing with Myc 1 and 2 for Max binding. Consistent with this, Myc3 is expressed in a distinct population of precursor cells committed for nerve and gland cell differentiation, although it is not expressed in the fully differentiated progeny.     

MAJOR COMMENTS

1. The EMSAs in Fig. 3 purport to show dose-dependent DNA binding by Myc3-Max heterodimers. The way I interpret these data, the authors mixed together equimolar amounts of recombinant proteins to generate the heterodimers.  However, I do not see any control experiments with purified individual Myc3 and Max proteins.  If Myc3 behaves like other Myc proteins, it should neither homodimerize nor bind DNA.  In contrast, Max should both homodimerize and bind DNA. Therefore it’s not clear whether the binding the authors show is attributable to My3-Max, Max-Max or some combination of the two. Using proteins of different sizes would allow the authoirs to distinguish Myc3-Max heterodimers and Max-Max homodimers

2. The lack of a TAD and its Myc boxes in Myc3 would lead one to predict that it would have little to no transforming activity. Indeed, the studies presented in Fig. 4A support this prediction. The two most important constructs are the second and third from the top.  The former, which contain the N-terminus of Myc3 and the dimerization domain of v-myc is essentially transformation-dead. This is expected given that the construct lacks a TAD and its Myc boxes. In contrast, attaching the N-terminus of v-myc to the dimerization domain of Myc3 is strongly transforming.  The simplest way to explain this is that, no matter how well Myc3  dimerizes, it will never be efficient at transforming unless it contains the essential N-terminal regions, i.e. Myc boxes within the TAD.  It might also lead one to predict that Myc3 would be a potent dominant negative inhibitor of Myc transformation that would compete for Max and exclude more potent Myc isoforms. These results also suggest that the lower level transformation seen when the N-terminus of v-Myc is fused to the dimerization domains of Myc1 or Myc2 are due to them being less efficient in forming heterodimers with Max. These conclusions are supported by the structure-based studies shown in Figs. 5 and 6. However, proof of this would best be done by again performing EMSAs with recombinant DNA binding domains of Max, v-myc and each of the 4 Hydra Mycs. Perhaps competition experiments between Max and different Hydra Mycs of different would allow one to directly compare their heterodimerization potentials and identify different Myc-Max complexes based on heterodimer-DNA size differences. Alternatively surface plasmon resonance might be a better way of looking at this as the authors suggest in the Discussion. These direct studies would confirm the structure-based predictions that the efficiency of Max dimerization is greater for Myc3 than for Myc1 or Myc2 and at least equivalent to that of v-myc.

MINOR COMMENTS

Lines 515-516: “Like many other oncogenes, myc was originally isolated from a transforming chicken retrovirus .

This should be re-stated so as to be more historically correct by saying that c-myc, the cellular oncogene, was originally isolated by virtue of its homology to its avian retroviral transforming counterpart, v-myc.

Lines 550-552: “Surprisingly, the C-terminal dimerization and DNA binding domain of Myc3 offers the highest structural conservation in comparison to Myc1 or Myc2, which is manifested in a transforming potential comparable to that of vertebrate Myc orthologues”.  This is a somewhat disingenuous conclusion. In actuality, there is nothing “oncogenic” about the Myc3 dimerization domain.  Rather, it is the ability to efficiently dimerize with Max (presumably with high affinity) that allows any TAD that is attached to it to more potently transform

Author Response

This is an interesting study that examines the role of one of 4 Myc proteins from the freshwater cnidarian polyp Hydra, namely Myc 3.  Myc 1 and Myc 2 are closely related the c-Myc of vertebrate, whereas Myc 3 is highly divergent and lacks most of the N-terminal transactivation domain of other Myc proteins and v-myc. Moreover, either because of the lack of the TAD or evolutionary divergence, it contains none of  “Myc boxes” that are found in other Myc proteins. Myc3 does dimerize strongl with Max and can bind E boxes in vitro. The Myc3 expression pattern in Hydra suggests a distinct role in cell differentiation that may involve Myc3 competing with Myc 1 and 2 for Max binding. Consistent with this, Myc3 is expressed in a distinct population of precursor cells committed for nerve and gland cell differentiation, although it is not expressed in the fully differentiated progeny.

We thank the referee for this constructive review. Below are our specific responses to the comments.

MAJOR COMMENTS

  1. The EMSAs in Fig. 3 purport to show dose-dependent DNA binding by Myc3-Max heterodimers. The way I interpret these data, the authors mixed together equimolar amounts of recombinant proteins to generate the heterodimers.  However, I do not see any control experiments with purified individual Myc3 and Max proteins.  If Myc3 behaves like other Myc proteins, it should neither homodimerize nor bind DNA.  In contrast, Max should both homodimerize and bind DNA. Therefore it’s not clear whether the binding the authors show is attributable to My3-Max, Max-Max or some combination of the two. Using proteins of different sizes would allow the authoirs to distinguish Myc3-Max heterodimers and Max-Max homodimers

We agree with this concern and have added additional data showing that Myc3 requires Max heterodimerization for efficient DNA binding. Figure A7 (formerly A6) already showed that Myc3 homodimers only poorly bind to DNA. We have now added shift assays with a direct comparison of Myc3:Myc3, Myc3:Max, and Max:Max dimers showing, as assumed by the referee, that also Max homodimers efficiently bind to DNA in contrast to Myc3 homodimers. Due to the different sizes of the individual protein-dimer/DNA complexes it can be deduced that the complex shown in Figure 3 must be a Myc3:Max heterodimer.

  1. The lack of a TAD and its Myc boxes in Myc3 would lead one to predict that it would have little to no transforming activity. Indeed, the studies presented in Fig. 4A support this prediction. The two most important constructs are the second and third from the top.  The former, which contain the N-terminus of Myc3 and the dimerization domain of v-myc is essentially transformation-dead. This is expected given that the construct lacks a TAD and its Myc boxes. In contrast, attaching the N-terminus of v-myc to the dimerization domain of Myc3 is strongly transforming.  The simplest way to explain this is that, no matter how well Myc3  dimerizes, it will never be efficient at transforming unless it contains the essential N-terminal regions, i.e. Myc boxes within the TAD.  It might also lead one to predict that Myc3 would be a potent dominant negative inhibitor of Myc transformation that would compete for Max and exclude more potent Myc isoforms. These results also suggest that the lower level transformation seen when the N-terminus of v-Myc is fused to the dimerization domains of Myc1 or Myc2 are due to them being less efficient in forming heterodimers with Max. These conclusions are supported by the structure-based studies shown in Figs. 5 and 6. However, proof of this would best be done by again performing EMSAs with recombinant DNA binding domains of Max, v-myc and each of the 4 Hydra Mycs.

The results from the molecular modeling analyses (cf. Figures 5, 6) suggest that Myc3/Max heterodimers have a higher stability compared to Myc2/Max or Myc1/Max dimers. However, we did not observe a significant lower Kd value of the Myc3/Max dimer compared to Myc2/Max or Myc1/Max dimers. We pointed out that a possible reason might be that the gel-based EMSA technique only allows estimation but not precise determination of dissociation constants from protein-DNA complexes. If the individual Hydra Myc proteins vary in these parameters as suggested by structural modeling, the differences may not be recorded by using classical protein-DNA detection tools. Gel-free alternatives such as surface plasmon resonance platforms or microscale thermophoresis could be employed thereby also testing Hydra Myc4 to precisely quantify Hydra Myc/Max dimerization and DNA interaction in free solution. However, we feel that these detailed analyses require a separate study also because in this paper only Myc3 is characterized but not Myc4. In fact, we plan to analyze Myc4 in a follow-up study where we also intend to investigate its DNA binding properties.

Perhaps competition experiments between Max and different Hydra Mycs of different would allow one to directly compare their heterodimerization potentials and identify different Myc-Max complexes based on heterodimer-DNA size differences. Alternatively surface plasmon resonance might be a better way of looking at this as the authors suggest in the Discussion. These direct studies would confirm the structure-based predictions that the efficiency of Max dimerization is greater for Myc3 than for Myc1 or Myc2 and at least equivalent to that of v-myc.

We agree that additional methods to analyze DNA are required (see above). Such experiments are planned where the binding properties of all four Hydra Myc proteins could be tested and compared to each other. However, we feel that this type of investigation is beyond the scope of the present manuscript, which deals exclusively with Hydra Myc3 but not Myc4.

MINOR COMMENTS

Lines 515-516: “Like many other oncogenes, myc was originally isolated from a transforming chicken retrovirus .”

This should be re-stated so as to be more historically correct by saying that c-myc, the cellular oncogene, was originally isolated by virtue of its homology to its avian retroviral transforming counterpart, v-myc.

The statement was modified accordingly also describing the origin of c-myc as the cellular counterpart of v-myc, which was transduced during retroviral infection by avian leukosis virus.

Lines 550-552: “Surprisingly, the C-terminal dimerization and DNA binding domain of Myc3 offers the highest structural conservation in comparison to Myc1 or Myc2, which is manifested in a transforming potential comparable to that of vertebrate Myc orthologues”.  This is a somewhat disingenuous conclusion. In actuality, there is nothing “oncogenic” about the Myc3 dimerization domain.  Rather, it is the ability to efficiently dimerize with Max (presumably with high affinity) that allows any TAD that is attached to it to more potently transform

This is correct. Therefore, this sentence was adapted to point out that the Myc C-terminus just serves as a surface for dimerization with Max and subsequent DNA binding, which is a prerequisite to position the oncogenic Myc transactivation domain to the relevant Myc target gene promoter regions.

Reviewer 3 Report

This paper presents two essentially unrelated pieces of work on the Myc3 gene of hydra. One is about the function of Myc3 in development. A hypothetical model is presented, based on single cell sequencing performed in previous studies. The second part of this investigation deals with the oncogenicity of Myc3. Myc3 can be expressed with RCAS in QEF. In these cells, Myc3 is not oncogenic. However, a chimeric protein consisting of an N-terminal portion derived from cMyc and the C-terminal Myc3 shows high oncogenicity. This activity is ascribed to the enhanced stability of the dimer formed by the chimeric protein and the Max protein. Mutation studies carried out in a heterologous system are in accord with this suggestion.

Overall, this is an interesting manuscript. Its major weakness lies in the combination of two unrelated observations that still need to be followed up in depth.

Specific comments:

·      The opening sentences of the introduction are about the history of Myc. That history is treated in a very cavalier manner, citing just three reviews. Yet, the ultimate authority on the origin of Myc sits right at the University of Innsbruck. The authors should consult with him, get a refresher in Myc history and then refer to the original work.

·      Introduction: Instead of that boiler-plate re-hash about Myc and cancer, why not provide more specifics on Myc3? Is the monomer disordered in solution? What is known about the N-terminal portion? Have the authors run any of the numerous motif and domain prediction programs on Myc3? Is Myc3 phosphorylated? What is its cellular localization? Which other proteins are seen in a Myc3 pull-down?

·      The terms “conserved” and “conservation” are frequently used, but it is never clear against which standard this conservation is measured. A theoretical leucine zipper or what?

·      There is a commercially available small molecule Myc inhibitor, KJ-Pyr-9. It would probably bind to the Myc3-MAX dimer. The authors should use it in the QEF work and in hydra. It would provide more functionally relevant information than the descriptive UMAP data.

·      Myc3 can be expressed with RCAS in QEF. An RNA seq analysis comparing QEF and QEF expressing Myc3 could be informative.

Author Response

This paper presents two essentially unrelated pieces of work on the Myc3 gene of hydra. One is about the function of Myc3 in development. A hypothetical model is presented, based on single cell sequencing performed in previous studies. The second part of this investigation deals with the oncogenicity of Myc3. Myc3 can be expressed with RCAS in QEF. In these cells, Myc3 is not oncogenic. However, a chimeric protein consisting of an N-terminal portion derived from cMyc and the C-terminal Myc3 shows high oncogenicity. This activity is ascribed to the enhanced stability of the dimer formed by the chimeric protein and the Max protein. Mutation studies carried out in a heterologous system are in accord with this suggestion.

Overall, this is an interesting manuscript. Its major weakness lies in the combination of two unrelated observations that still need to be followed up in depth.

We thank the referee for the positive comment. We agree that both observations should be studied in more detail to elucidate the underlying molecular mechanisms. However, in this first paper, we describe structure, expression and biochemical properties of this novel Myc isoform, and we tried to focus on the capacity of Myc3 to form hetero-dimers with Max and its role in oncogenic transformation. We felt that these findings can be combined into one manuscript to give an initial overview of Myc3, because the biochemical and the expression data in Hydra strongly suggest that Myc3-Max interaction plays a role in cell transformation and differentiation. We tried to express this link in our working model (Figure 8).

Specific comments:

  • The opening sentences of the introduction are about the history of Myc. That history is treated in a very cavalier manner, citing just three reviews. Yet, the ultimate authority on the origin of Myc sits right at the University of Innsbruck. The authors should consult with him, get a refresher in Myc history and then refer to the original work.

We apologize for the impression that the historic description of Myc may have been neglected too much. Following the recommendation, the introduction has now been adapted accordingly thereby also referring to Myc history in more detail including original citations.

  • Introduction: Instead of that boiler-plate re-hash about Myc and cancer, why not provide more specifics on Myc3? Is the monomer disordered in solution? What is known about the N-terminal portion? Have the authors run any of the numerous motif and domain prediction programs on Myc3? Is Myc3 phosphorylated? What is its cellular localization? Which other proteins are seen in a Myc3 pull-down?

So far, Myc3 has not been published and apart from our studies nothing is known publicly about this novel Myc isoform. We have now performed a motif and domain prediction analysis and mention the outcome in the results section. Concerning the proposed cell biological and biochemical experiments with regard to subcellular localization, phosphorylation, and protein-protein interactions apart from Max, we would rather like to investigate this in a follow-up study.

  • The terms “conserved” and “conservation” are frequently used, but it is never clear against which standard this conservation is measured. A theoretical leucine zipper or what?

The term conservation is based on a certain threshold of sequence identity between homologous nucleic acid or protein sequences between different species. We have explained this now in the paper after the first usage of this term.

  • There is a commercially available small molecule Myc inhibitor, KJ-Pyr-9. It would probably bind to the Myc3-MAX dimer. The authors should use it in the QEF work and in hydra. It would provide more functionally relevant information than the descriptive UMAP data.

We have applied this inhibitor in previous studies (Hart et al., 2014; Raffeiner et al., 2014). Again, testing this compound or other Myc inhibitors, which became available in the meantime, on Hydra Myc proteins could be performed in a separate study. The presented paper already has eight figures in the main part and ten illustrations in the supplemental material.

  • Myc3 can be expressed with RCAS in QEF. An RNA seq analysis comparing QEF and QEF expressing Myc3 could be informative.

This is an exciting proposition since the quail transcriptome is now available (Marasco et al., 2020), which could be also applied to study the v-myc oncogene in the original avian cell system where it was initially discovered. However, we feel that this interesting project is beyond the scope of the presented manuscript.

Round 2

Reviewer 1 Report

My concerns have been addressed.

Reviewer 2 Report

The authors have adequately addressed my concerns and comments

Reviewer 3 Report

no comments